# Analysis of Relation between Brainwave Activity and Reaction Time of Short-Haul Pilots Based on EEG Data

**DOI:** 10.3390/s23146470

**Published:** 2023-07-17

**Authors:** Bartosz Binias, Dariusz Myszor, Sandra Binias, Krzysztof A. Cyran

**Affiliations:** 1Department of Data Science and Engineering, Silesian University of Technology, Akademicka 16, 44-100 Gliwice, Poland; 2Department of Algorithmics and Software, Silesian University of Technology, Akademicka 16, 44-100 Gliwice, Poland; dariusz.myszor@polsl.pl; 3Laboratory of Sequencing, Nencki Institute of Experimental Biology of the Polish Academy of Sciences, 02-093 Warsaw, Poland; s.binias@nencki.edu.pl; 4Department of Graphics, Computer Vision and Digital Systems, Silesian University of Technology, Akademicka 16, 44-100 Gliwice, Poland; krzysztof.cyran@polsl.pl

**Keywords:** aircraft control human factors, cognitive workload, data mining, electroencephalography, fatigue, safety

## Abstract

The purpose of this research is to examine and assess the relation between a pilot’s concentration and reaction time with specific brain activity during short-haul flights. Participants took part in one-hour long flight sessions performed on the FNPT II class flight simulator. Subjects were instructed to respond to unexpected events that occurred during the flight. The brainwaves of each participant were recorded with the Emotiv EPOC+ Scientific Contextual EEG device. The majority of participants showed a statistically significant, positive correlation between Theta Power in the frontal lobe and response time. Additionally, most subjects exhibited statistically significant, positive correlations between band-power and reaction times in the Theta range for the temporal and parietal lobes. Statistically significant event-related changes (ERC) were observed for the majority of subjects in the frontal lobe for Theta frequencies, Beta waves in the frontal lobe and in all lobes for the Gamma band. Notably, significant ERC was also observed for Theta and Beta frequencies in the temporal and occipital Lobes, Alpha waves in the frontal, parietal and occipital lobes for most participants. A difference in brain activity patterns was observed, depending on the performance in time-restricted tasks.

## 1. Introduction

Safety is an important aspect of the modern airline industry. Among numerous factors that impact the proper execution of the flight process, pilot performance is one of the most crucial. Conducted surveys point out that pilot fatigue was recognized as a probable cause of 21–23% major aviation accidents [1,2]. The importance of fatigue counteraction is confirmed by aviation organizations like National Transportation Safety Board which over the past 25 years created more than 200 recommendations related to pilot tiredness [3]. Noteworthy, based on studies conducted in 1980 and 2017 we can conclude that despite the efforts put into limiting the harmful effects caused by pilot tiredness, the overall tendency does not appear to have changed significantly over the years [4]. Moreover, a survey conducted on the group of short-haul pilots points out that over 75% of them claimed that they were subject to significant fatigue [5] and that over 70% of corporate pilots claimed to have experienced micro-sleep episodes during various phases of the flight [6]. This state is associated with decreased responsiveness to external stimuli [7] and deterioration in cognitive tasks [8].

The neurophysiological evaluation of tiredness in aircraft pilots is crucial for ensuring flight safety, optimizing pilot performance, complying with regulations, implementing effective fatigue risk management, and promoting pilot well-being. It allows for the detection and management of fatigue-related risks, contributing to safer and more efficient aviation operations. Neurophysiological evaluation involves the assessment of various physiological measures and processes within the nervous system. One method used to achieve this is brain signal analysis, which specifically focuses on analyzing the electrical activity generated by the brain. A very popular method of brain signal recording is *electroencephalography* (EEG) [9,10,11]. In this method, measurement sensors, known as electrodes, are placed in specific locations over the scalp. These electrodes record the changes in electric potential caused by neural activity. Since EEG signals have low amplitudes (typically ranging from 0.5 to 100μV) and bandwidth mostly located below 100 Hz, they are highly susceptible to noise and artifact disturbances. Therefore, the common practice involves using differential measurement configurations, such as mono- or bipolar, to reduce data contamination. In order to ensure the reproducibility of the conducted EEG research the electrodes are typically located and labeled according to a universally accepted standard [10]. This standard is designed to provide optimal coverage of the functional areas of the brain (i.e., Brodmann’s Areas). One example of such a standard is the *10-20* system, which is commonly used for electrode montage. Variations of this system, such as *10-10* and *10-5*, have also been developed [12]. As a result of EEG measurement, an *electroencephalogram* (EEG) is obtained. In the EEG signal some characteristic frequency bands can be distinguished. These are referred to as: Delta (below 3 Hz), Theta (3–7 Hz), Alpha (8–12 Hz), Beta (13–29 Hz) and Gamma (over 30 Hz) [11,13,14]. It is worth mentioning that the frequency limits of specific waves are conventional as there is no proper way of determining their exact values.

The EEG-based neurophysiological evaluation of tiredness or fatigue in vehicle drivers and aircraft pilots has been the subject of numerous research papers in recent years, highlighting its significance in understanding and addressing pilot fatigue in aviation.

The relationships between EEG, ECG, and peripheral physiological signals to assess driver cognitive load and driver performance during simulated driving tasks are explored in [15]. A real-time approach for evaluating driver cognitive load using single-channel EEG is presented in [16]. The work proposes a feature extraction and classification framework to monitor and assess driver workload levels during on-road driving experiments. Another study focuses on driver fatigue detection using EEG signals and machine learning techniques [17]. It explores the classification of different fatigue levels based on EEG features extracted during driving experiments. In a study by Aricò et al. [18], neurophysiological measures of pilots, including EEG, were examined to assess variations in mental workload across different flight phases/varying flight durations. The research explores the changes in cognitive states and workload demands during takeoff, cruise, and landing. The study by Borghini et al. [19] specifically focuses on the EEG-based evaluation of pilot mental workload during simulated flight scenarios. EEG features are assessed to detect variations in workload levels and investigate their correlation with pilot performance. A neurophysiological assessment of mental workload in simulated piloted flights using Bayesian machine learning is presented in [20]. EEG data are utilized to estimate and classify different levels of mental workload experienced by pilots. The neurophysiological evaluation of mental workload in aircraft carrier approach and landing tasks using EEG is presented in [21]. The work explores the relationship between EEG measures and pilot workload during complex aviation operations. Another interesting study focuses on the quantitative assessment of mental workload in aircraft carrier approach and landing using single-channel EEG [22]. An EEG-based workload index to measure and analyze pilot cognitive states during critical flight operations is proposed. The mental workload in aircraft carrier pilots using EEG-based functional connectivity analysis is evaluated in [23]. Changes in functional brain network connectivity patterns and their relation to pilot workload during carrier operations are examined.

The goals of this research are related to an assessment of the relation between the bandpower of the pilot’s EEG signal and reaction time with specific brain activity during short-haul flights. The achieved results lead to a better understanding of the cognitive activity underlying the decision-making processes that occur during time-sensitive events and require immediate reaction. Proper identification of factors leading to a decreased performance in such actions is critical. Potentially, it could drastically increase the safety of flights. Additionally, early detection of changes in mental activity that are associated with drowsiness, micro-sleeps and lack of focus could allow applying adequate precautionary measures in advance. Finally, obtaining the ability to predict the pilot’s intended action before the execution of voluntary movements required to perform it, could be utilized in various neuroadaptive technology solutions [24]. Among many appealing examples of such systems are the cognitive cockpit systems [25,26,27]. Until now, multiple studies have reported on the relationship between specific frequency bands (such as Alpha, Beta, Theta and Gamma) in the EEG signal and cognitive processes associated with attention, memory, and cognitive control. These cognitive processes can have an impact on reaction time in various tasks ([28,29,30,31]). However, these studies do not specifically focus on pilots or short-haul pilots but provide general insights into the relationship between brainwave activity and reaction time. The main novelty of the proposed research lies in its specific focus on the correlation between brainwave activity and reaction time, particularly in the context of short-haul flights. The analysis and identification of differences in brain activity patterns for different reaction times (slow, medium, and fast), further highlights the novelty of the study. These findings contribute new insights to the field and provide evidence of a previously unexplored relationship between brainwave activity and reaction time specifically in the context of short-haul pilots.

## 2. Materials and Methods

### 2.1. Description of Experiment

To evaluate the performance of the pilots during short-haul flights, a set of experiments was conducted during which participants were flying in simulated conditions of a professional FNPT II class simulator. The objective of the experimentation phase was to measure the activity of the human brain, during the simulated session of short-haul flights, with the autopilot activated. Visual cues were randomly displayed on the main screen of the simulator during the experiment sessions.

Participants in the study were selected from a group of individuals aged between 20 and 35 years old. All participants claimed that they were well-rested prior to the session, and all of them gave consent for the utilization of the experimental outcomes for scientific research purposes. Subjects had no prior training, nor experience related to aircraft or flight simulators. During the experimentation phase 8 people were examined. Data were gathered at the same time of the day (around 12 a.m.) and it was ensured that no external factors influenced the participants. Each session lasted around 1 hour and was preceded by a short introduction and installation of an EEG device on the pilot’s head. Participants were seated in the FNPT II class simulator. They were instructed to act as regular pilots, focusing on observing cockpit instruments and scanning the surroundings of the simulated plane. They were specifically instructed to maintain awareness and to remain focused in order to be able to react instantly to the occurrence of visual cue events. When a visual cue was displayed on the screen, participants were instructed to press a designated button immediately. To minimize the time required for reacting to the visual cue, the button was placed in a location that did not require any additional movements of the pilot’s body besides their fingers.

In order to maintain consistency between consecutive experimental sessions, a simulated flight from Frankfurt to London was recorded and the same section of the flight was presented to every participant. Both the terrain over which the flight took place as well as cockpit instruments were registered. During this flight autopilot was activated. The flight took place at an average altitude of 6000 feet. To simulate a flight with the autopilot activated, the recorded material excluded the take-off and landing phases. In addition, the entire flight, presented to the participants, took place over the land. Notably, engine sounds were also generated in the cockpit.

Visual cues were randomly displayed with normal distribution characterized by a mean of 2.5 min and a standard deviation of 1 min. The introduction of variance was intended to prevent the human brain from habituating to regular patterns. Additionally, the distribution of visual cues in time was consistent for each pilot. The visual cue was represented by a solid, red-colored box that overlaps 75% of the main simulator screen responsible for displaying the terrain. The consent of the bioethics committee was obtained, which allowed the conducting of such experiments.

### 2.2. Flight Simulators

During experimentation sessions, a professional Flight Navigational Procedure Training II (FNPT II) class simulator was utilized (Figure 1). The simulator was built by SoftekSim company (Riga, Latvia), and is based on Lockheed Martin Prepar3D software (software version number Prepar3D 1.4) that reflects the Cessna 172 RG plane model. It has successfully passed QTG tests. The flight simulator features a fully enclosed full-size cockpit that faithfully reproduces the internals of the Cessna 172 RG, including a glass cockpit. The simulator offers a 180-degree panoramic view of the environment, that is generated by three projectors. To create an immersive environment, the simulator is located in a specially designated space (Virtual Flight Laboratory at the Silesian University of Technology). The laboratory has no windows and is designed with black walls to ensure that no external stimuli can reach the pilot during the simulation.

### 2.3. Emotiv Epoc+ Headset

The EEG data used in this research were recorded using the Emotiv EPOC+ Headset. This device utilizes a sequential sampling method with a rate of 128 SPS and provides signals of 14 bit (1LSB=0.51μV). The device incorporates a built-in digital 5-th order Sinc filter and notch filters at 50 Hz and 60 Hz, to enhance the quality of the recorded signal. The recorded signals have a useful bandwidth ranging from 0.16 to 43 Hz [32]. The Emotiv EPOC+ Headset is compatible with the international 10-10 electrode montage system and provides access to 14 EEG channels. The available electrodes placements are: AF3, F7, F3, FC5, T7, P7, O1, O2, P8, T8, FC6, F4, F8, AF4, with references at the P3/P4 locations (monopolar) [32]. The placement of EPOC+ electrodes with regards to the 10−10 configuration is depicted in Figure 2 [33]. The Emotiv EPOC+ headset uses semi-dry electrodes. A semi-dry electrode is a type of electrode used in electroencephalography that requires minimal or no conductive gel for signal acquisition. The design aims to improve user-friendliness, comfort, and ease of application compared to traditional wet electrodes [34,35].

The cerebral cortex of the human brain can be divided into four major lobes: Frontal, Parietal, Occipital and Temporal. This anatomical classification aligns with the functional classification of different areas of the brain [36]. Presented in Table 1 are the cerebral lobes that correspond to the specific electrodes available in the Emotiv Epoc+ Headset [33,37].

The Emotiv EPOC+ Headset system provides an affordable and practical solution for basic scientific research. However, it has been observed that the recorded EEG data can be susceptible to artifacts caused by muscle movements, such as limb actions, head repositioning, or blinking. In order to reduce the impact of such artifacts on the experimental results, special precautions were undertaken. All participants were seated in a comfortable position and instructed to minimize their movements during data recording. Additionally, the reference and event segments were manually evaluated for the presence of artifacts. Trials that were assessed to contain excessive contamination were excluded from the analysis.

Presented in Figure 2 are the locations of the EPOC+ electrodes with relation to the *10-10* configuration [12,33].

### 2.4. Digital Processing of EEG Signals

In this research, EEG signals were analyzed independently, focusing on specific frequency ranges corresponding to different brain waves such as Theta, Alpha, Beta, and Gamma. To achieve this, a method for spectral filtering of these signals was employed. For the purpose of bandpass filtering of EEG data, a zero-phase (non-delaying) filter was applied during offline processing. This is usually implemented by applying a recursive filter to the original signal both forwards and backwards in time.

Let $x\in R^{M}$ be a recorded, discrete signal consisting of length *M* and *h* be the impulse response of the recursive filter. The output $v\in R^{M}$ of filtering operation performed on *x* is calculated as in (Equation 1).
(1)v=h∗x. If x(i) (i=1,…,M) denotes a discrete sample o *x*, then the operation of flipping the signal can be defined as in (Equation 2) [38].
(2)$$\forall i\in Z,i<M:flip\left(x(i)\right)=x\left(M-i\right).$$ The *flip* operator reverses the order of samples of a discrete signal *x* [38]. Considering the above definitions the output of forward–backward filter $y\in R^{M}$ can be calculated as presented in (Equation 3) [38].
(3)$$y=flip\left(h*flip\left(h*x\right)\right).$$ In this research, a Kaiser Window Finite Impulse Response (FIR) band-pass filter constructed of 466 coefficients was used. Due to their linear-phase characteristics, FIR filters are well-suited for biomedical signal processing applications. However, their disadvantage is manifested by non-negligible delays that are introduced to the data as a result of their use. The approach that was implemented in this research allows for benefiting from the advantages of FIR filters, while at the same time, eliminating problems related to phase delays in offline processing. Besides spectral division of EEG signals, bandpass filtering additionally improves the key characteristics such as Signal-to-Noise Ratio (SNR) and removes distortions caused by electrical line drift and high-frequency noise.

### 2.5. Description of Data Features

The band power features used for the analysis of brain activity during the experiment were extracted from spectrally filtered signals, individually for each measurement channel. Since the mean value of bandpass filtered EEG signal is close to 0 its power is equivalent to its variance. To normalize the distribution of calculated features a logarithm operation is commonly applied [39]. A logarithm of the variance of the signal’s amplitude calculated during a specific time interval is a very popular feature used for the description of EEG signal’s power in specific frequency band [39]. The frequency ranges used to represent specific brain waves for the purpose of the analysis are presented below:Theta waves: 3–7 HzAlpha waves: 8–12 HzBeta waves: 13–29 HzGamma waves: 30–69 Hz

The choice of the time interval for calculating the band power features is an important consideration in this research. Two segments of the signal were taken into account for each event. The *reference* segment begins 2.5 s before the visual cue indicating the start of each new event and ends finishes 0.5 s before the marker appears. This segment serves as a baseline for determining the level of change in mental activity with the appearance of visual stimuli. It also represents the idle brain activity. The *activity* segment consisted of all samples within 4.5 s time window from the beginning of an event. The length of this segment was selected based on the reaction times achieved by subjects during the experiment. It was observed that 98% of all reaction times did not exceed 4.5 s.

Visual representation of the segments used in this research with regard to the appearance of the visual cue is shown in Figure 3.

Although calculated in the same manner and with respect to the assumption of ERD/ERS, a metric used in this research, will be referred to as Event-Related Change and calculated on the basis of band power of the *activity* and *reference* segments as presented in (Equation 4).
(4)$$ERC=\frac{log\left(var(\mathrm{activity})\right)-log\left(var(\mathrm{reference})\right)}{log\left(var(\mathrm{reference})\right)}$$

## 3. Results of the Analysis

### 3.1. Relation between Response Time and Experiments Duration

A trend analysis was conducted on the experimental data, to determine the existence of an association between the reaction time of each subject to the appearance of the visual cue and the duration of the experiment. For this purpose, a linear function was fitted to data with the timestamp of the appearance of the visual cue serving as function input and time of reaction as output. The coefficients of the function were estimated with a simple linear regression method.

To assess the significance of estimated linear regression coefficients an F-test was performed. The null hypothesis stated that all non-constant coefficients of the regression equation are zero. The alternate hypothesis stated that at least one of the non-constant coefficients in the regression equation is not equal to zero. Since the timestamp of the event was the only explanatory variable, the rejection of the null hypothesis indicated the non-zero value of the slope coefficient. The proposed trend analysis approach was based on the analysis of the sign of that coefficient. For statistically significant results the positive values would reveal a tendency of the subject’s response time to increase over the time of an experiment while negative values would suggest the opposite. The *p*-values of calculated F-statistics for each subject are shown below:Subject 1: 0.52685Subject 2: 0.72606Subject 3: 0.07392Subject 4: 0.1870Subject 5: 0.060716Subject 6: 0.19262Subject 7: 0.09430Subject 8: 0.94558

The *p*-values calculated for each subject exceeded 0.05 indicating that the test did not achieve a reasonable confidence level. Therefore, there is no reason to reject the null hypothesis. On that basis, it can be determined that no linear trend changes in response times related to the duration of performed experiments were present. This observation is consistent with the visual inspection of the experimental data.

### 3.2. Relation between Experiment Duration and General Brain Activity

The goal of this stage of the analysis was to determine whether there was any overall increase or decrease in the subject’s brainwave-related band power in a specific cerebral location, over the course of the experimental session.

For the purpose of this analysis, an approach similar to that proposed for the evaluation of the trend in response times was implemented. A linear regression equation was fitted with the time of an event as an explanatory variable. As a dependent variable, a smoothed logarithm of signal power in individual bands calculated from the *reference* segments preceding events was used. The data were smoothed using a Simple Moving Average Filter (SMA) with a length of 128 (1-s time window). The selection of *reference* segment for this purpose is justified by the fact that these time segments represent idle brain activity. During that time segments subjects were not performing any other mental activity other than focusing on the upcoming event. Therefore, the use of the *reference* segment is perfectly suited for determining whether any changes in awareness or fatigue occur over the time of the experiment. To evaluate the significance of the linear regression slope coefficient an F-test with p=0.01 was performed.

The dominant trend direction for each subject was determined based on partial trends of all 8 powers courses calculated on the basis of all electrodes belonging to a given lobe. If over 75% of electrode trends had the same sign of trend direction, that direction was assigned as dominant, indicating either power increase (denoted as +) or decrease (denoted as −). If over 75% of estimated slope coefficients were recognized as statistically insignificant (i.e., null hypothesis that they are equal to 0 could not be rejected) it was assumed that no significant increase or decrease in power occurred. If none of the above could be determined, meaning that no trend behavior was common for at least 75% of the frontal lobe electrodes, it was assumed that changes were inconsistent (denoted as NC for *Non Consistent*). Table 2, Table 3, Table 4 and Table 5 present the summaries of the subject’s dominant trends of power changes in Theta, Alpha and Beta waves with respect to the frontal, temporal, parietal and occipital lobes. Examination of obtained results reveals that for the majority of subjects, statistically significant changes in any of the bandwidths were not observed in that region.

### 3.3. Correlation between EEG Power of Brainwaves and Response Times of Subjects

The Pearson correlation coefficients between the amount of brain power related to specific brainwaves and specific response times were calculated individually for each subject, in order to determine the existence and nature of the relation between these factors. Specifically, the correlation between response time to an event and logarithmic Theta and Beta powers in the *activity* segment that was present directly after the presentation of the cue were examined independently. The correlation values calculated for each electrode in each band-power are presented in Appendix B.

It was observed that the preponderance of the correlation coefficients are positive. Therefore, further analysis is focused on evaluating the statistical significance of the positive nature of observed correlations. Statistical tests for the significance of the positive correlation coefficient with α=0.05 (for one-tailed critical values) were performed on the obtained results. The acceptance of the null hypothesis was unequivocal with the statement that the population correlation coefficient equals 0. Therefore, in such a situation no association between data could be claimed. Rejection of the null hypothesis would reveal that a positive correlation could exist. The positive, and statistically significant, correlation could potentially imply that the longer the time required for the subject’s reaction, the greater the mental workload generated in the specific frequency range. Table 6, Table 7, Table 8 and Table 9 present the summaries of subjects who exhibit a statistically significant correlation (at a confidence level α=0.05) between Theta, Alpha, Beta and Gamma band power in specific cerebral lobes.

A statistically significant, positive correlation between Theta Power in the frontal lobe and response time was reported for the preponderance of the participants (6 out of 8). Additionally, the majority of subjects (5 out of 8) exhibited statistically significant, positive correlations between band-power and reaction times in the Theta range for the temporal and parietal lobes. Similarly, over half of the subjects showed positive correlations reported in the Frontal Lobe for the Beta range. Half of the subjects had positive correlations for the Theta range in the occipital lobes, the Alpha range in the frontal, parietal and occipital lobes and the Beta range in the parietal lobes.

### 3.4. Analysis of Frequency-Specific Event-Related Changes

One particularly interesting aspect of this research was to analyze and determine whether any change in signal power could be observed between signals occurring before and after the same trial event. In order to infer this information, Student’s two-tailed *t*-test with critical value of α=0.05 was performed to assess whether the mean value of all ERC, measured for the subject during the session, was significantly different from zero. The null hypothesis stated that no significant Event Related Changes occurred during the session and the alternative hypothesis said that the mean of all ERC during the session was non-zero. Since ERC values can be both positive and negative, there is a possibility that high relative changes will result in a mean of zero (i.e., in the case of alternating ERC signs), despite the fact that absolute ERC value may indicate a strong change in band power. Nevertheless, such a situation would imply that there is no consistent pattern of brain power change (increase or decrease) that could be generalized into applicable conclusions. Therefore, rejection of such cases by the proposed statistical test will be desired.

Table 10, Table 11, Table 12 and Table 13 present subjects who achieved non-zero mean in specific band power and cerebral lobes.

Statistically significant ERC could be observed for almost all of the subjects in the frontal lobe for Theta frequencies (7 out of 8). The majority of subjects (6 out of 8) exhibit significant changes in EEG signal power for Beta waves in the frontal lobe and in the frontal as well as in the temporal lobes for the Gamma band. It is worth noting, that for most (5 out of 8) of the participants, significant ERC could be observed for Theta and Beta frequencies in the temporal and occipital lobes, for Alpha in the frontal lobe and Gamma in the parietal and occipital lobes.

### 3.5. Analysis of Event-Related Changes Activity Maps

An investigation of the differences in brain activity patterns corresponding to the reaction times has been performed. The reaction times were first assigned a rank of either slow, medium or fast. The ranking is based on the tertile to which each reaction time belonged. Therefore, reaction times from the first tertile were the fastest for a given individual and assigned fast rank. The third tertile was associated with the slowest reaction times and, consequently, assigned to the slow group. The remaining reaction times (second tertile) were ranked as medium. The grouping was performed independently for each subject to avoid bias arising from the individual characteristics of the participants. Differences in natural predispositions could have led to incorrect interpretations of the results (i.e., medium reaction times for one subject might be fast for another one). Once the reaction times were ranked, the ERCs calculated in Theta, Alpha, Beta and Gamma EEG frequencies were averaged within the rank and visualized.

Presented in Appendix A are activity maps of brain activity visualized for different wavelengths. The ERCs are grouped by reaction time tertiles according to the procedure described in this section. In addition to maps for individual subjects Figure A1, Figure A2, Figure A3, Figure A4, Figure A5, Figure A6, Figure A7 and Figure A8, an averaged activity map for all subjects has been included in Figure A9. The activity maps were generated with MNE-Python package [40].

It can be observed that brain activity patterns of all subjects for slow reaction times are characterized by an increase in the ERC in the left hemisphere of the temporal lobe. This applies to almost all frequency ranges analyzed in this work. The most significant, observed activity patterns are described below

ERC increase in the left hemisphere of the temporal lobe in Theta frequencies was found for all subjects, except Subject 2.ERC increase in the left hemisphere of the temporal lobe (often slightly overlapping with the occipital area) in Alpha frequencies was found for subjects 1, 4, 6, 7 and 8. Similar activity but shifted to the temporal lobe was observed for Subjects 2 and 5.ERC increase in the left hemisphere of the temporal lobe in Beta frequencies was found for Subjects 2, 3, 4, 5, 6 and 8.ERC increase in the frontal lobe in Gamma frequencies was found for Subjects 1, 3, 4, 5 and 6.

These observations are further supported by the analysis of the average activity patterns for slow rank presented in Figure A9. Temporal activity in Theta and Alpha waves and frontal activation in the Gamma range are especially clearly defined.

Analysis of maps of brain activity regarding medium reaction times allowed for the identification of patterns that are specific to groups of subjects. Alpha ERC increase in the left area of the temporal lobe could be observed for almost all subjects. The only exception to this was found for Subject 7 (Figure A7) that exhibited only a decrease in ERC in the right area of the frontal lobe. Additionally, four smaller cohorts of subjects could be observed that shared similar activity patterns for medium reaction times:ERC increase in the left area of the temporal lobe in Alpha frequencies could be observed for almost all subjects.ERC increase in the left hemisphere of the occipital lobe in Beta frequencies was found for Subjects 1, 2, 5, and 8.ERC increase in the right hemisphere of the frontal lobe in Beta frequencies was found for Subjects 2, 4, 7 and 8. Subject 6 had similar activity in the central part of that lobe.ERC increase in the occipital lobe in Gamma frequencies was found for Subjects 1, 3, 4, 5, and 8.ERC increase in the frontal lobe in Gamma frequencies was found for Subjects 2, 3, 4, 6, 7 and 8.

Interestingly, apart from right frontal activity in the Beta range and no clear pattern in Theta, the observed activities are not visible in the averaged activity maps plot presented in Figure A9.

In general, the right hemisphere seems to be the most involved part of the brain in actions related to fast reactions. The activity is mainly visible in the frontal lobe as increases in the ERC. The most outstanding observations have been described below:ERC increase in the occipital lobe in Theta frequencies was found for Subjects 1, 2, 3, 4, 5, 6 and 8.ERC increase in the right hemisphere of the frontal lobe in Alpha frequencies was found for Subjects 1, 2, 3, and 5.ERC increase in the right hemisphere of the frontal lobe in Beta frequencies was found for Subjects 1, 3, 4, 5, 7 and 8.ERC increase in the right part of the frontal lobe in Gamma frequencies was found for Subjects 1, 3, 4, 5, 7, and 8.

Apart from right frontal activity in Beta and central frontal in Gamma, patterns described above are not clearly visible on the average activity patterns presented in Figure A9.

## 4. Discussion

### 4.1. Positivity of Correlation between Reaction Time and EEG Power

A positive correlation between Theta Power and reaction time was observed for all lobes in the presented research. Similar findings were observed in the cortical regions of the brain and reported in [41]. The study used high-resolution EEG mapping techniques to examine cortical activation related to working memory tasks. While the specific brain regions were not explicitly mentioned in the summary, the study provides insights into the relationship between Theta Power and reaction time within the broader cortical network involved in working memory processes. While not covering the subject of aircraft fatigue, the findings of the mentioned study are compliant with the conclusions drawn in this work.

Similarly, a positive correlation has been observed in participants of this study in Beta Power in frontal as well as parietal lobes. The correlation between EEG oscillations, including Beta Power, and cognitive and memory performance was discussed in [28]. Although the specific focus of that work is not solely on reaction time, it provides valuable insights into the relationship between Beta Power and cognitive processes. According to this study, Beta oscillations in the EEG signal are typically associated with active cognitive processing, including attention, alertness, and motor planning. Increased Beta Power has been observed during tasks that require focused attention, concentration, and the execution of motor actions. In the context of reaction time, higher Beta Power may reflect heightened cognitive engagement and readiness to respond to stimuli.

Lower Alpha Power has been linked to increased cognitive activity, attention, and information processing [28]. Thus, a positive correlation between Alpha Power and reaction time could suggest that decreased Alpha Power reflects enhanced cognitive readiness and faster response capabilities. This is well aligned with the findings presented in this work, reporting the statistically significant, positive correlation between Alpha Power and reaction time in frontal, parietal or occipital lobes

The positivity of correlations may be attributed to the greater amount of power required in order to leave the state of attentive cue awaiting (i.e., subject became temporarily less focused on the task). Another explanation could relate this phenomenon to the case-specific, harder and more demanding, process of determining the appropriate reaction to be performed (i.e., stronger brain activity was required). Both of these factors lead to an elongation of time devoted to the processing and association of visual cues, which directly translates to a delayed or slower response to the event. Concluding, more Theta-related or Beta-related activity is interpreted as related to more demanding brain processing, which justifies slower response time to stimuli. It must be noted that the aforementioned factors are not mutually exclusive. Additionally, they are not related to the subject’s general level of fatigue, drowsiness or concentration since other analyses of these data have shown that no significant overall decrease or increase in these states was present. Therefore, it is concluded that these factors are of temporal nature and occur locally around the time of a visual cue presentation. For further discussion, let us assume that higher demand for neuronal activity indicates that deciding on the proper action is more difficult from a cognitive perspective. Based on such an assumption, it can be speculated that higher reaction times were, at least partially, a result of brain’s inability to efficiently process the visual cue. This can be further supported by the presence of Non-Zero ERC of signal power in the occipital lobe. Such changes are interpreted as statistically significant differences between the signal’s energy before and after the presentation of the visual queue. A significant elevation in energy might be a result of either its low state before the event or increased demand after the queue. The first scenario might be attributed to lower concentration. This is especially true in the case of Beta waves, since these waves are mostly associated with tasks requiring attention and concentration.

### 4.2. Detection of Changes in Brain Activity

In neurosciences, it is assumed that the appearance of a stimulus can induce changes in the neuronal activity time-locked to an event, known as event-related potentials (ERP) [42]. The event-related phenomena can additionally represent frequency-specific changes in the EEG which are manifested as either an increase or decrease in signal’s band power. This phenomenon is called Event-Related Desynchronization or Synchronization (ERD/ERS) [42]. A metric based on the ERD/ERS was selected to assess the changes in mental activity after the appearance of the stimulus. In their work, Pfurtscheller and da Silva precisely formulated the assumptions behind the ERD: ‘the term ERD is only meaningful if the baseline measured some seconds before the event represents a rhythmics seen as a clear peak in the power spectrum’ [42].

Detection of frequency-specific event-related changes from EEG signals was not a main focus of the conducted research. Its purpose was to evaluate the potential of using such data as part of a Man–Machine Interaction solution for cognitive cockpit systems. Therefore, the nature of these changes was not taken under examination. Instead, the focus was placed on determining whether such behavior is present and displays any characteristics related to frequency and location that can be generalized to a greater group of subjects. Changes in EEG power in Theta frequencies were observed in almost all subjects in all of the lobes. Although the specific focus is not solely on reacting to visual cues, the study discusses the analysis of EEG power in the Theta frequency range as a measure of driver fatigue [43]. The results indicate that changes in Theta power can be indicative of cognitive states and alertness levels. While the article does not directly address the relationship between Theta power changes and reaction to visual cues, it demonstrates the relevance of Theta power analysis in assessing cognitive states and fatigue.

Alpha power decreases with attentional focus and cognitive load, while increasing during relaxed wakefulness or when attention is disengaged. Resting-state Alpha activity is linked to individual differences in cognitive abilities. Alpha oscillations play a crucial role in attention, perception, and cognitive control, making them a valuable topic of investigation in neuroscience research [28]. The role of Alpha oscillations during visual perception and attention is very important. Specifically, if the modulation of power in this range is present during visual-processing tasks, including the reaction to visual cues. This might well explain the findings presented in this article, where many of the participants had statistically significant changes of the Alpha power in the frontal lobe when reacting to the unexpected event.

Beta waves have been found to be related to the reaction to unexpected events. Studies have shown that Beta power can exhibit changes or modulations during the processing of unexpected stimuli or events [44]. Changes in this activity have also been reported in this study for the majority of the subjects.

Gamma oscillations are believed to play a crucial role in various cognitive processes and sensory perception [45]. They have been commonly associated with the processing of unexpected events [46]. When encountering unexpected or surprising stimuli, the brain often exhibits enhanced Gamma power and synchronization. Interestingly, consistent changes in Gamma power were observed across lobes for the majority of the participants. Achieved results prove that performing a reaction-based task is detectable on the basis of EEG recordings. Therefore, such signals have the potential to be used in cognitive cockpit applications. However, it is important to note that the specific roles and mechanisms of Gamma oscillations in response to unexpected events may vary across different experimental paradigms and brain regions. Further research is needed to elucidate the precise contributions of Gamma oscillations in the context of unexpected event processing.

### 4.3. ERC Activity Patterns for Different Groups of Reaction Times

An interesting observation can be drawn when analyzing the differences between the most common brain activity patterns for slow, medium and fast reaction time groups. Left temporal increase of ERC in both Theta and Alpha waves seems to be most prevalent for slow reaction times. With the improvement of reaction times (medium range), it can be observed that activity in Theta is silenced. At the same time, temporal activity in Alpha remains clearly visible. Finally, for the fastest reaction times, Alpha activity shifts towards Beta and Gamma in the right hemisphere of the frontal lobe. Additionally, activation in the occipital lobe becomes visible in the Theta range. Based on these observations it can be claimed that both, the greater involvement of the right hemisphere in the frontal lobe and lesser temporal activity in the Alpha range, combined with greater activation of the Occipital Lobe in the Theta range, contributes to the faster reaction times of the participants. This is perhaps the most significant finding of the analysis of brain activity maps conducted within this research. Another interesting observation can be drawn for Gamma frequencies. For the slowest reaction times, Gamma activity is mostly prevalent in the central area of the frontal lobe. While this remains the case for the reaction times ranked as medium in Section 3.5, Gamma wave energy starts increasing in the occipital lobe as well. For the fastest reaction times, the Gamma activity remains strongly expressed in the frontal lobe. However, it can be observed that the activity patterns become shifted towards the right hemisphere. Based on these observations, it can be further concluded that greater activation of the right hemisphere in the frontal area of the cerebrum is associated with faster reaction times.

The most important role of the frontal lobe is processing tasks associated with planning, motivation, short-term memory and attention. This also involves the discrimination between events and the assessment of consequences associated with performed actions. Additionally, the frontal lobe plays a major role in voluntary movement planning and control [36,37]. The occipital lobe is responsible for visual-processing actions such as focus and identification of stimuli, motion perception, visuospatial orientation and colour differentiation. It also takes part in coordinating motor actions in response to external stimuli [36,37]. Processing of somatic sensation as well as integrating sensory information from various parts of the body is generally assigned to the parietal lobe [36,37]. The temporal lobe is mostly involved in auditory processing both on a low and a high level (i.e., language recognition). Additionally, there are areas of this lobe that are associated with interpreting the visual stimuli and establishing object recognition [36,37].

### 4.4. Impact of the Duration of the Session on Performance

Statistical analysis of estimated trend directions of the relation between the response time of a particular subject and the duration of experimental sessions did not reveal any significant, non-constant tendencies. This observation complies well with the visual inspection of the data. In general, it was noted that all deviations in response time of each subject did not exhibit any time dependence. The reaction times were calculated as a difference between the time of the subjects’ reaction to the visual cue and the time when the cue appeared. Therefore, higher values of this statistic should be interpreted as a slower response to the stimuli. It was assumed that the time required for the signal to be transferred from the keyboard to the logging device is negligible. This remark leads to the conclusion that a one-hour-long experimental session was not sufficiently long enough in order to downgrade or in any other way influence the performance of subjects in the posed task.

It was generally observed that as the experimental sessions progressed, brain power of Theta, Alpha, Beta and Gamma waves maintained a steady level in the frontal, temporal, parietal and occipital areas of the cerebrum. This may lead to the conclusion that the proposed experiment did not induce states of either drowsiness or fatigue among subjects. Analogous observations were made during the analysis of the response times. This should be attributed to the duration of each experimental session. To summarize both of these conclusions, it must be stated that according to this research, one-hour-long sessions of simulated flight did not contribute in any way to the long-term changes of either band power in any part of the cerebrum or decreased response time of subjects. Therefore, such short sessions can be considered relatively safe in terms of mental workload, fatigue and drowsiness of pilots.

### 4.5. Relation to Previous Work

This work complements previously conducted preliminary research during which the effect of the extended duration of performing monotonous manoeuvres on drowsiness and mental fatigue was examined [47]. In addition, previous research has demonstrated that it is possible to predict reaction times on the basis of EEG data [48]. Results of another study confirmed the possibility of using EEG-based BCI systems in cognitive cockpit solutions [49]. This has been achieved by presenting an accurate machine learning model allowing for the discrimination between states of brain activity related to idle but focused anticipation of visual cue and reaction to this cue. The purpose of this work was to provide a detailed assessment and analysis of the nature of the mechanisms that enabled the successful utilization of EEG data for the prediction and classification of participants’ activity patterns. Signal energy of the Lower Gamma band (32–36 Hz) in the frontal lobe (AF3 and F8 electrodes) was the most commonly selected feature for the purpose of prediction of pilot’s reaction time based on EEG signals in a recent study [48]. The main finding regarding positive correlation in that range in frontal lobe complements and helps to explain the significance and effectiveness of these features selected in said work.

### 4.6. Impact of Volume Conduction and Electrode Placement

Because of the phenomena known as the *volume conduction*, fields originating from distant sources of bioelectrical activity are diffused and propagated through brain tissues. As a result, they reach multiple electrodes and mix with the signals produced by local sources. Because of this EEG signals are characterized by very low spatial resolution. It has been shown that the sources within a 3 cm radius of each scalp electrode contribute only partly to the measured signal [50]. The effects of source overlapping are sometimes corrected with the use of *spatial filtering* methods [51,52]. However, this research follows a methodology applied in our previous work where spatial filtering has been omitted [48]. The benefit of this approach is the consistency of the results obtained over a series of analyses performed on the same data. Additionally, during the experiment, it was observed that EPOC+ electrodes were not always positioned precisely at the 10−10 locations assigned to them. This discrepancy may be attributed to the construction limitations of the EPOC+ system, which does not allow for significant adjustments of electrode placement. The inaccuracies in electrode positioning could arise due to variations in the shape and size of the subject’s skull. Additionally, some further misplacements could result from adjusting the electrode position for the best quality of the recorded signal. However, it must be noted that said misplacements were mostly of a subtle nature and had never led to the displacement of the electrode outside of the cerebral lobe it was originally assigned to. Considering the aforementioned observations, it was decided to focus on the more general-activity-related lobes rather than on the specific locations of electrodes.

## 5. Conclusions

The presented research findings demonstrate a significant, positive correlation between Theta Power in the frontal lobe and response time, indicating that increased Theta activity in this region is associated with longer response times. Moreover, a majority of participants exhibited significant, positive correlations between band-power and reaction times in the Theta range for the temporal and parietal lobes, suggesting the involvement of these regions in cognitive processing related to response time. Additionally, positive correlations were found in over half of the subjects in the frontal lobe for the Beta range, indicating a potential role of Beta activity in modulating response times. Furthermore, half of the subjects exhibited positive correlations in the Theta range for the occipital lobes, Alpha range for the frontal, parietal, and occipital lobes, and Beta range for the parietal lobes, implying the involvement of these brain regions and frequency bands in the processing and coordination of response times. These findings contribute to our understanding of the neurophysiological mechanisms underlying response time variability and highlight the importance of investigating specific brain regions and frequency bands in relation to cognitive performance. Further research is warranted to elucidate the precise mechanisms and functional implications of these correlations. Positive correlations between Theta power in the frontal lobe and the subject’s response time were found for most participants. Such results indicate that the higher the power of the EEG signal was in the frontal lobe after the visual cue appeared, the longer it took for the subjects to react appropriately. This may be attributed to findings reported in the literature concerning the relation between memory workload during cognitive processing in this band. This finding is additionally supported by the commonly accepted association of the frontal lobe to the processing of various tasks (including those requiring attention) and discrimination between events. In the research, this can be interpreted as assessing whether the observed marker corresponds to the assumed visual cue and on that basis deciding whether to press the button or not.

Event-related changes in the EEG signal proved to be an effective metric. It can be successfully used to assess the existence of changes in signal power caused by the appearance of visual stimuli. Since such signals are related to mental state, the ERC metric can be additionally considered as a meaningful descriptor of changes in brain activity. In this work, statistically significant ERC measures were observed for Theta frequencies in the frontal lobe for the majority of subjects, indicating a consistent neural synchronization in this region during task performance. Additionally, significant changes in EEG signal power were found for Beta waves in the frontal lobe, as well as in the frontal and temporal lobes for the Gamma band, suggesting the involvement of these frequency bands and brain regions in cognitive processing. Moreover, it is noteworthy that significant ERC measures were observed in the temporal and occipital Lobes for both Theta and Beta frequencies in the majority of participants, indicating the presence of neural synchronization in these regions during task-related activities. Furthermore, Alpha frequency demonstrated significant ERC in the frontal lobe, while the Gamma band exhibited significant ERC in the parietal and occipital lobes for most participants. These findings shed light on the neural connectivity patterns across different brain regions and frequency bands, providing valuable insights into the underlying mechanisms of cognitive processing. Further investigations are warranted to explore the functional significance of these observed coherence patterns and their potential implications in cognitive tasks.

Analysis of the common activity patterns presented in Section 3.5 leads to an interesting observation that the involvement of certain parts of the cerebrum results in different performance in time-restricted tasks. This is an intriguing subject that, if further analyzed, might help to better understand the nature of the components of fast and slow reflexes and reactions.

The fact that, apart from some minor exceptions, the mental activity and reaction times of almost all subjects did not show any significant signs of progressing tiredness, drowsiness or mental fatigue was compliant with subjective self-assessment of their own state. This leads to an important observation, that one-hour-long sessions of flight attentive monitoring interrupted by occasional fast response demanding tasks do not affect mental states related to fatigue and tiredness of participants. Therefore, for tasks like this, or in some way similar, such duration can be considered as safe. Conducting research with longer experimental sessions and more diversified stimuli (i.e., auditory) are necessary steps that will enable further investigation of this subject.

This work focuses on the analysis of participants’ brain power, reaction times and their relation. Authors try to provide an interpretation of the results in neurocognitive and physiological contexts based on the known theory, most recent research and the nature of the experiment. However, it must be kept in mind that such conclusions cannot be accurately validated as it is impossible to assess the difficulty regarding specific repetition of the task for the subject or whether the participant felt unfocused shortly before the event occurred. Trying to assess this during the experiment (i.e., through a survey or questionnaire) would constitute a disturbance and greatly affect the outcome. At the same time, asking the subject about individual trials at the end of the experiment would not be reliable.

## Figures and Tables

**Figure 1 sensors-23-06470-f001:**
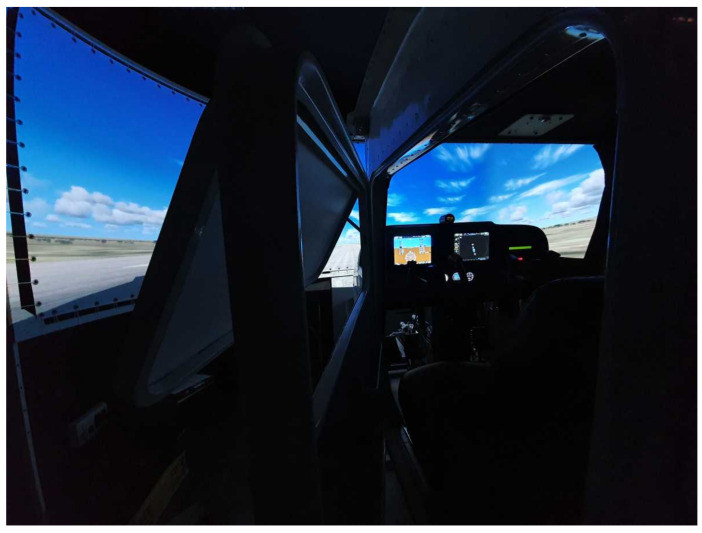
FNPT II Cessna flight simulator employed during research.

**Figure 2 sensors-23-06470-f002:**
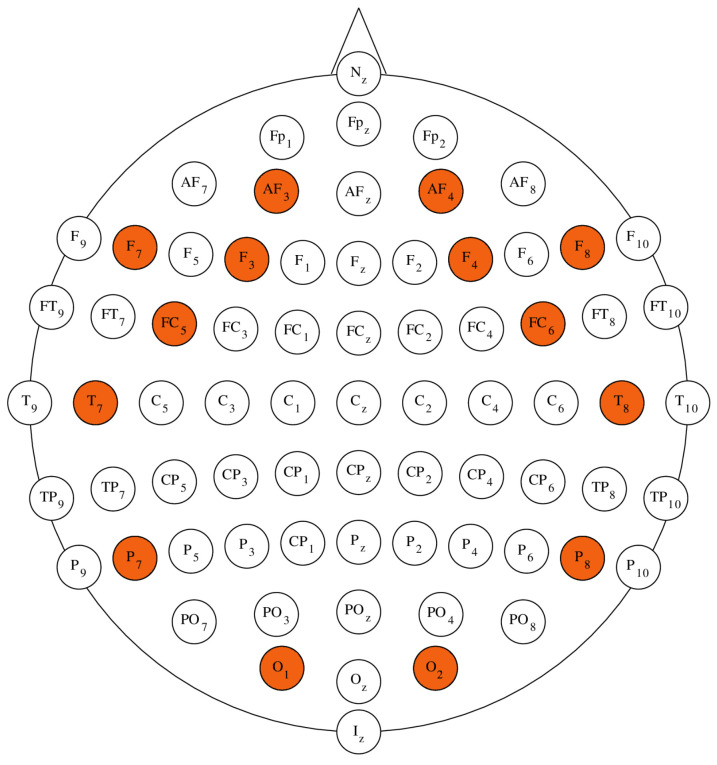
Positions of electrodes in the standard *10-10* electrode montage system (own source procedurally generated based on [12]). Red color denotes electrode locations available in the Emotiv Epoc+ Headset.

**Figure 3 sensors-23-06470-f003:**
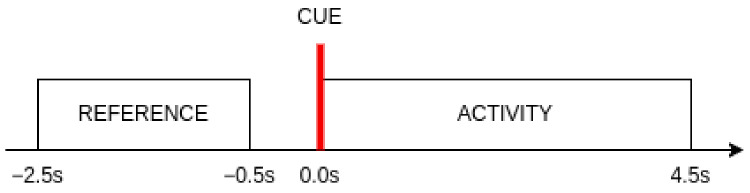
Visual representation of the segments used in this research with relation to the appearance of the visual cue.

**Table 1 sensors-23-06470-t001:** Cerebral Lobes Corresponding to the Locations of Emotiv Epoc+ Headset Electrodes [33,37].

Electrodes	Lobe
AF3, AF4	Frontal
F7, F8	Frontal
F3, F4	Frontal
FC5, FC6	Frontal
T7, T8	Temporal
P7, P8	Parietal
O1, O2	Occipital

**Table 2 sensors-23-06470-t002:** Summary of Dominant Trend Directions in Frontal Lobe of All Subjects (+—Power Increase, =—No Significant Changes, NC—Inconsistent Changes).

Band	Subjects
Theta	1(=), 2(=), 3(=), 4(=), 5(=), 6(=), 7(=), 8(=)
Alpha	1(=), 2(=), 3(=), 4(=), 5(+), 6(=), 7(=), 8(=)
Beta	1(=), 2(=), 3(=), 4(=), 5(=), 6(=), 7(=), 8(=)
Gamma	1(=), 2(=), 3(NC), 4(=), 5(=), 6(=), 7(=), 8(=)

**Table 3 sensors-23-06470-t003:** Summary of Dominant Trend Directions in the Temporal Lobe of All Subjects (=—No Significant Changes, NC—Inconsistent Changes).

Band	Subjects
Theta	1(=), 2(=), 3(=), 4(=), 5(NC), 6(=), 7(=), 8(NC)
Alpha	1(=), 2(=), 3(=), 4(=), 5(NC), 6(=), 7(=), 8(=)
Beta	1(=), 2(=), 3(NC), 4(=), 5(=), 6(=), 7(=), 8(=)
Gamma	1(=), 2(=), 3(NC), 4(=), 5(=), 6(=), 7(=), 8(NC)

**Table 4 sensors-23-06470-t004:** Summary of Dominant Trend Directions in the Parietal Lobe of All Subjects (+—Power Increase, =—No Significant Changes, NC—Inconsistent Changes).

Band	Subjects
Theta	1(=), 2(=), 3(=), 4(=), 5(NC), 6(=), 7(=), 8(=)
Alpha	1(=), 2(=), 3(=), 4(=), 5(NC), 6(=), 7(=), 8(=)
Beta	1(=), 2(=), 3(=), 4(=), 5(=), 6(=), 7(=), 8(=)
Gamma	1(=), 2(=), 3(+), 4(=), 5(=), 6(=), 7(=), 8(=)

**Table 5 sensors-23-06470-t005:** Summary of Dominant Trend Directions in the Occipital Lobe of All Subjects (+—Power Increase, =—No Significant Changes).

Band	Subjects
Theta	1(=), 2(=), 3(=), 4(=), 5(+), 6(=), 7(=), 8(=),
Alpha	1(=), 2(=), 3(=), 4(=), 5(+), 6(=), 7(=), 8(=),
Beta	1(=), 2(=), 3(=), 4(=), 5(=), 6(=), 7(=), 8(=),
Gamma	1(=), 2(=), 3(+), 4(=), 5(=), 6(=), 7(=), 8(=),

**Table 6 sensors-23-06470-t006:** Subjects with Significant (α=0.05), Positive Correlations of Theta Wave Power in Specific Cerebral Lobes.

Lobe	Subjects
Frontal	1, 2, 3, 4, 5, 7
Temporal	2, 4, 5, 6, 8
Parietal	2, 3, 4, 5, 7
Occipital	2, 4, 5, 6

**Table 7 sensors-23-06470-t007:** Subjects with Significant (α=0.05), Positive Correlations of Alpha Wave Power in Specific Cerebral Lobes.

Lobe	Subjects
Frontal	1, 2, 5, 8
Temporal	2, 4, 5
Parietal	2, 5, 6, 7
Occipital	1, 2, 5, 6

**Table 8 sensors-23-06470-t008:** Subjects with Significant (α=0.05), Positive Correlations of Beta Wave Power in Specific Cerebral Lobes.

Lobe	Subjects
Frontal	1, 2, 3, 4, 5
Temporal	1, 2, 4
Parietal	1, 2, 5, 7
Occipital	2, 5, 6

**Table 9 sensors-23-06470-t009:** Subjects with Significant (α=0.05), Positive Correlations of Gamma Wave Power in Specific Cerebral Lobes.

Lobe	Subjects
Frontal	1, 2
Temporal	1, 2
Parietal	1, 2,
Occipital	1, 2, 5

**Table 10 sensors-23-06470-t010:** Subjects with Statistically Significant (α=0.05) Non-Zero Event Related Changes in the Frontal Lobe.

Band	Subjects
Theta	1, 2, 4, 5, 6, 7, 8
Alpha	1, 2, 4, 5, 8
Beta	2, 4, 5, 6, 7, 8
Gamma	2, 4, 5, 6, 7, 8

**Table 11 sensors-23-06470-t011:** Subjects with Statistically Significant (α=0.05) Non-Zero Event Related Changes in the Temporal Lobe.

Band	Subjects
Theta	2, 4, 5, 6, 8
Alpha	4, 5, 6
Beta	2, 4, 6, 7, 8
Gamma	2, 4, 5, 6, 7, 8

**Table 12 sensors-23-06470-t012:** Subjects with Statistically Significant (α=0.05) Non-Zero Event Related Changes in the Parietal Lobe.

Band	Subjects
Theta	2, 4, 5, 7
Alpha	5
Beta	2, 4, 5, 6
Gamma	4, 5, 6, 7, 8

**Table 13 sensors-23-06470-t013:** Subjects with Statistically Significant (α=0.05) Non-Zero Event Related Changes in the Occipital Lobe.

Band	Subjects
Theta	2, 4, 5, 8
Alpha	
Beta	2, 4, 8
Gamma	2, 4, 5, 6, 8

## Data Availability

Not applicable.

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
