# Peer review of "Analysis of Relation between Brainwave Activity and Reaction Time of Short-Haul Pilots Based on EEG Data"

_sensors, 2023, doi:10.3390/s23146470_

Round 1

Reviewer 1 Report

The authors examine and assess the relation between the pilot’s concentration and reaction time with specific brain activity during short-haul flights. Although being interesting, I find that there are some major issues with the paper that require addressing prior to this being considered for publication in this journal. I have identified the main points for consideration below:

1.     This manuscript has some spelling typos, style errors and grammatical errors. Please carefully check the whole manuscript and correct them.

2.     The novelty of this study should be clarified in the revised manuscript.

3.     EEG electrodes is essential for EEG acquisition. Semidry electrode has been a typical electrode for biopotential recording. So, I recommend that the author add the description of semidry electrode in the introduction section. In addition, some recent related references are recommended to be cited, for example, J. Neural Eng. 18 (2021) 046016; Journal of Neural Engineering 20 (2023) 026017.

4.     How many subjects participated in the EEG signal recording?

 Moderate editing of English language required.

Author Response

Dear Reviewer

We are extremely grateful for all the remarks, suggestions and time spent reviewing our manuscript. 

Based on the reviews we have reorganized the article by introducing following changes: 

  • we have combined Material and method into one chapter with divisions
  • we removed the description of the well-known data on the origin of the EEG from the introduction
  • we presented more detailed literature review and comparison to the relevant research, in particular covering the studies on vehicle drivers, for pilots with varying flight durations, and for aircraft carrier pilots
  • we mproved the Discussions section both by moving relevant parts of the manuscript to this section (according to Reviewers suggestions) and by discussing the results with relation to findings from the literature.

We have addressed all the points from the review in a way described below:

  1.     We have now made a best effort to review and correct or spelling errors and typos. We assure that once the review stage is finalized we will subject the whole manuscript to the professional proofreading and correction process. Considering that at this stage, manuscript might still undergo a major modification, we are postponing this process. We apologize for all the mistake that we have missed.
  2.     We hope that the novelty of the article is now better highlighted in the Introduction, Discussion and Conclusion sections.
  3.     As suggested we have added the description of semidry electrode in the “2.3. Emotiv Epoc+ Headset” section with reference taken from Journal of Neural Engineering 
  4.     In total, only 8 subject participated in this experiment. We hope that this information is now better presented in the “2.1. Description of experiment” section.

We hope that introduced changes meet your expectations and look forward to receiving further comments and remarks.

Sincerely,

Authors

Reviewer 2 Report

The title and aim of the manuscript sound intriguing. But having become acquainted with the material, I recommend paying attention to a line of points during processing.

The general remark, but not critical. The authors present EEG data for 8 subjects. Small groups in neurophysiology are possible, but either with clinical pathology, or with the use of other methods (psychological, etc.), or with the use of complex analysis methods (for example, deep convolutional neural networks, etc.). Therefore, as an option, I recommend that the authors change the goal to a research protocol that has less requirements.

1.   INTRODUCTION. It is necessary to prove the novelty of the work. It makes no sense to describe the well-known data on the origin of the EEG. The scientific literature contains a sufficient number of works related to the neurophysiological evaluation of operators. Scientific evidence is available for vehicle drivers, for pilots with varying flight durations, and even for aircraft carrier pilots. And, of course, for astronauts. All this is well described. Such literature data would raise the value of this paper both in the introduction and in the discussion.

Speaking about EEG methodology, especially the authors discuss not the traditional 10-20 system, but 10-10 and 10-5, it is better to cite methodological scientific sources – the International Federation of Clinical Neurophysiology (or Clinical Neurophysiology European Societies).

Move the last block from the introduction (lines 94-117) to Methods.

2.     Material and methods. It is better to combine into one Chapter with divisions. Brief description - trained subjects or random people. EEG - monopolar or bipolar? Fig. 1 should be moved here. Tab. 1 and further in an appendix - P7, P8 - these are parietal sites (Parietal). 

The contents of lines 176 - 207 are not for Methods. It will look better in the Discussion.

Lines 253 - 266 - to the Discussion, as well as subchapter 3.2.

Values of the table 2 is the minimum for the text, since there is no reliability. You can just put it in the text.

295 - 300 is a repetition of what was above. 

3. DISCUSSION. As such, there is no discussion. There is no comparison in this chapter with similar results from other studies that I mentioned above. The authors again describe their results without discussion as such. Reading the discussion, one forgets about the main task of the study on the research of pilots. It is also required for this Chapter. 

4. The abstract should be like a mini article. Conclusion - the main results for the task. 

5. Appendix. One or two Figs. can be transferred to the results. The rest don't make sense. Figure captions are not correct, because these are not just topographic maps, but ERC maps. The indicators should indicate the quantitative values of the colors.

In tables - the results are very heterogeneous and the tables are practically not described in the text. "Values in bold represent statistically significant correlations" - usually in neurophysiology it is customary * to denote differences with decoding in the instructions under the table. 

6. References. 34 - commercial brochure. For a scientific journal such a reference is not accepted. 

Author Response

Dear Reviewer

We are extremely grateful for all the remarks, suggestions and time spent reviewing our manuscript. We have addressed all the points from the review in a way described below:

  1. We have improved the Introduction section in a following way: 
  • proved the novelty of the work by presenting more detailed comparison to the relevant research, in particular covering the studies on vehicle drivers, for pilots with varying flight durations, and for aircraft carrier pilots,
  • removed the description of the well-known data on the origin of the EEG
  • We have changed the citation for the electrode montage systems to “The standardized EEG electrode array of the IFCN” (https://doi.org/10.1016/j.clinph.2017.06.254)
  • We moved the last block from the introduction (lines 94-117) to Methods.
  1.     As suggested we have combined Material and method into one chapter with divisions. Required additional information (untrained subjects, monopolar recording) was added into appropriate subsections.

Other changes:

  • Fig. 1 moved to  Material and methods
  • We have corrected the reference to P7, P8 electrodes as parietal sites (Parietal). This required redoing the analysis with Parietal Lobe also included and correcting the conclusions
  • The contents of lines 176 - 207 were moved from Methods to the Discussion.
  • Lines 253 - 266 - to the Discussion, as well as subchapter 3.2.
  • Table 2 has been turned into a text.
  • Part from lines 295 - 300 was rephrased to focus on the purpose of the paragraph
  1. We have reworked the Discussion section, both by moving relevant parts of the manuscript to this section (according to Reviewers suggestions) and by discussing the results with relation to findings from the literature.
  2. We have reworked the abstract to better present the conclusions of the work
  3. We believe that the figures provided in the Appendix server as a reference and proof for the observations that were made for the differences in brain activity patterns for slow, medium and fast reaction times. In our opinion, without the figures it would be hard to present data that justifies our findings which we consider an important part of the article in an approachable way. If required, we are willing to move the figures either to “supplementary data” or remove them.

Figure captions have been corrected to ERC maps. Since our analysis was performed on normalized data we believed that current legend levels were more descriptive. We have however changed them to the range of [-1, 0, 1] and added the interpretation of the values to the description.

Considering that we have observed a great variety of the levels of raw data values presented on the legend, we believe that introducing these raw values could be potentially misleading. Especially, since we compare the subjective (normalized) activation levels of the subjects.

We have replaced the term “EEG Topographic map” with “ERC activity map”.

Tables with Pearson correlation values have been requested by other Reviewer and, if acceptable, we would prefer to keep them in the Appendix or move them to “Supplementary Data”. We have change the instruction to follow * convention

  1. References: We have replaced reference to commercial brochure with reference to “Validation of Emotiv EPOC+ for extracting ERP correlates of emotional face processing” (https://doi.org/10.1016/j.bbe.2018.06.006)

We hope that introduced changes meet your expectations and look forward to receiveing further comments and remarks.

Sincerely,

Authors

Reviewer 3 Report

Sensors-2443526

Analysis of Relation Between Brainwave Activity and Reaction Time of Short-Haul Pilots Based on EEG Data

 1. The authors require re-writing the abstract to show clearly the objective of this paper.

2. The innovation of the article is not appropriate.

3. The writing of the article is not appropriate so that it is difficult for the reader to understand it.

4. The block diagram that explains the proposed method should be added to the article

5. The proposed method is general, so it is necessary to provide more complete explanations in the conclusion section.

6. Comparison with other new references should be made and the superiority of the method should be stated

 Moderate editing of English language required

Author Response

Dear Reviewer

We are extremely grateful for all the remarks, suggestions and time spent reviewing our manuscript. We have addressed all the points from the review in a way described below:

  1. We have rewritten the abstract to better present the purpose and findings of the article.

  1. We hope that the novelty of the article is now better highlighted in the Introduction, Discussion and Conclusion sections.

  1. We have reorganized the article by introducing following changes: we have combined Material and method into one chapter with divisions, removed the description of the well-known data on the origin of the EEG, presented more detailed literature review and comparison to the relevant research, in particular covering the studies on vehicle drivers, for pilots with varying flight durations, and for aircraft carrier pilots, improved the Discussions section both by moving relevant parts of the manuscript to this section (according to Reviewers suggestions) and by discussing the results with relation to findings from the literature.

4-5. The main purpose of this article is to present results of the statistical analysis of the experimental data that has been gathered. We believe the data processing and analysis methods used in this work are not it’s main contribution and follow standard methodology used in EEG signal processing. If still required we are willing to include more details (i.e. block diagram) on the data processing pipeline used in this work. However, at this point we would prefer to limit this information to avoid this information drawing away the attention of readers from the main objectives of the research.

  1. Comparison with findings from other references has been added to in the Discussions section. All findings are now addressed and compared to related reports from the literature.

We hope that introduced changes meet your expectations and look forward to receiving further comments and remarks.

Sincerely,

Authors

Round 2

Reviewer 1 Report

1.     This manuscript has some spelling typos, style errors and grammatical errors. Please carefully check the whole manuscript and correct them.

2.     EEG electrodes is essential for EEG acquisition. Semidry electrode has been a typical electrode for biopotential recording. So, I recommend that the author add the description of semidry electrode in the introduction section. In addition, some recent related references are recommended to be cited, for example, Journal of Neural Engineering 20 (2023) 026017.

Moderate editing of English language required.

Author Response

Dear Reviewer

Thank you for your valuable input. 

We have corrected all spelling, style and grammatical errors. 

In addition, we have added citation of Li, Guangli, et al. "Polyvinyl alcohol/polyacrylamide double-network hydrogel-based semi-dry electrodes for robust electroencephalography recording at hairy scalp for noninvasive brain–computer interfaces." Journal of Neural Engineering 20.2 (2023): 026017. to the manuscript

We hope that introduced changes meet your expectations and look forward to receiving further comments and remarks.
Sincerely,
Authors

Reviewer 2 Report

The manuscript may be published in the present form.

Author Response

Dear Reviewer

We would like to, once again thank you and express our gratitude for the extremely valuable comments and suggestions. We believe you have helped us to greatly improve the quality of our research paper.

For this round, we have only corrected all spelling, style and grammatical errors. Additionally, we've include one, new citation to address suggestion of other Reviewer.

Sincerely,
Authors

Reviewer 3 Report

.

 Minor editing of English language required

Author Response

Dear Reviewer

Thank you for your valuable input. 

We have corrected all spelling, style and grammatical errors. 

In addition, we have included one, new citation as recommended by other Reviewer.

We hope that all introduced changes fulfil your expectations and that you will find our manuscript appropriate. We are looking forward to receiving further feedback.

Sincerely,
Authors